# Myofibroblasts: A key promoter of tumorigenesis following radiofrequency tumor ablation

**Marwan Moussa[1], David Mwin[1], Haixing Liao[2,3], M. Fatih Atac[1], Aurelia Markezana[4], Eithan Galun[4], S. Nahum Goldberg[1,2], Muneeb Ahmed[1]** *

**1** Department of Radiology, The Laboratory for Minimally Invasive Tumor Therapies, Beth Israel Deaconess Medical Center/Harvard Medical School, Boston, MA, United States of America, **2** Department of Radiology, Hadassah Hebrew University Hospital, Jerusalem, Israel, **3** The First Affiliated Hospital of Guangzhou Medical University, Guangzhou, China, **4** Goldyne Savad Institute of Gene Therapy, Hadassah Hebrew University Hospital, Jerusalem, Israel

* mahmed@bidmc.harvard.edu

**Data Availability Statement:** All relevant data are within the paper and its Supporting Information files.

## Abstract

Radiofrequency ablation (RFA) of intrahepatic tumors induces distant tumor growth through activation of interleukin 6/signal transducer and activator of transcription 3 (STAT3)/hepatocyte growth factor (HGF)/tyrosine-protein kinase Met (c-MET) pathway. Yet, the predominant cellular source still needs to be identified as specific roles of the many types of periablational infiltrating immune cells requires further clarification. Here we report the key role of activated myofibroblasts in RFA-induced tumorigenesis and successful pharmacologic blockade. Murine models simulating RF tumorigenic effects on a macrometastatic tumor and intrahepatic micrometastatic deposits after liver ablation and a macrometastatic tumor after kidney ablation were used. Immune assays of ablated normal parenchyma demonstrated significantly increased numbers of activated myofibroblasts in the periablational rim, as well as increased HGF levels, recruitment other cellular infiltrates; macrophages, dendritic cells and natural killer cells, HGF dependent growth factors; fibroblast growth factor-19 (FGF-19) and receptor of Vascular Endothelial Growth Factor-1 (VEGFR-1), and proliferative indices; Ki-67 and CD34 for microvascular density. Furthermore, macrometastatic models demonstrated accelerated distant tumor growth at 7d post-RFA while micrometastatic models demonstrated increased intrahepatic deposit size and number at 14 and 21 days post-RFA. Multi-day atorvastatin, a selective fibroblast inhibitor, inhibited RFA-induced HGF and downstream growth factors, cellular markers and proliferative indices. Specifically, atorvastatin treatment reduced cellular and proliferative indices to baseline levels in the micrometastatic models, however only partially in macrometastatic models. Furthermore, adjuvant atorvastatin completely inhibited accelerated growth of macrometastasis and negated increased micrometastatic intrahepatic burden. Thus, activated myofibroblasts drive RF-induced tumorigenesis at a cellular level via induction of the HGF/c-MET/STAT3 axis, and can be successfully pharmacologically suppressed.

**Funding:** This work is supported by National Cancer Institute (1R01CA197081-01A1), Israel Ministry of Science and Technology (3-12063), and Israel Science Foundation (1277/15). The funders had no role in study design, data collection and analysis, decision to publish, or preparation of the manuscript.

**Competing interests:** The authors have declared that no competing interests exist.

## Introduction

Image-guided thermal ablation is an established clinical treatment that can achieve outcomes equivalent to surgical resection in well-selected patients [1–5], particularly for tumors less than 3 cm [6, 7]. Thermal ablation is now being used widely given it's applicability to a growing cohort of non-surgical patients and a relatively lower economic cost [8–12]. While local treatment efficacy has continued to improve due to growing expertise and technological improvements, there is increasing evidence that effects from local ablation may result in unintentional downstream systemic side effects. Specifically, increased growth of distant untreated tumor following local ablation has been observed in both animal models [13–17] and with variable frequency in some clinical scenarios as unintended phenomena to both ablation [18, 19] and other interventional oncologic procedures [20–25].

Prior studies have linked post-ablation tumorigenesis to cytokinetic processes and accompanying inflammatory cellular infiltration that occur in the red peripheral zone surrounding an ablation. For liver ablation, Interlukin-6 (IL-6), hepatocyte growth factor and its high affinity receptor mesenchymal epithelial transition factor receptor (HGF/c-MET) axis [26–29] have been identified as major drivers of these effects. In HCC, IL-6 and the HGF/c-MET axis have been linked to increased angiogenesis through VEGF-1/A pathway, and proliferation through fibroblast growth factor-19 (FGF-19) [30–33]. Ablation-induced increase in IL-6 has been previously reported in clinical studies and in preclinical models, and successful blockade of this cytokine has curbed ablation induced distant tumor growth [14, 15, 17, 32, 34].

While specific cytokines have been identified as key promoters of post-ablation distant tumor growth and have been successfully blocked with pharmacologic therapy, the precise cellular sources of these cytokines have yet to be elucidated and potentially targeted. Accordingly, here we attempt to identify key cell populations driving post-ablation tumorigenesis. A growing body of literature implicates cancer-associated fibroblasts (CAFs) as a key cell population in promotion of tumorigenesis and proliferation pathways [31]. Given the prominence of fibroblasts in the periablational rim and their known potential to produce a host of growth factors including those found elevated post-RF ablation, we hypothesized that activated fibroblasts may play a key role in regulating unintentional distant tumor growth post-ablation. Moreover, we hypothesized that anti-fibroblast drugs such as atorvastatin may successfully attenuate tumorigenesis of distant metastasis. Accordingly, the purpose of this study was to characterize the role of myofibroblasts within the ablation zone on ablation-induced stimulation of distant tumors and micrometastatic disease in preclinical models, and to further study the use of an adjuvant anti-fibroblast agent, atorvastatin, on suppressing myofibroblast-driven ablation-induced distant tumor growth and metastatic implantation.

## Material and methods

### Overview

Animal studies were performed in accordance with protocols approved by the Institutional Animal Care Committees of Beth Israel Deaconess Medical Center, and Hadassah Hebrew University Medical Center, respectively. A total of 124 rats were implanted with a subcutaneous rat breast cancer cell line to characterize the role of myofibroblasts on both RFA-induced distant macrometastatic tumor growth and 64 mice were subject to intrahepatic implantation with two different murine colorectal cancer cell lines to characterize the role of micrometastatic tumor implantation and growth. The study was conducted over four phases.

## Phase I

Determination of the role of myofibroblasts in hepatic RFA-induced HGF upregulation, periablational cell trafficking and downstream growth pathways modulation. First, 48 non-tumor bearing Fischer rats were randomized into the following treatment groups: 1) sham treatment, 2) sham with multiple doses of atorvastatin, 3) RFA of the liver, and 4) RFA of the liver with multiple doses of atorvastatin. Animals were sacrificed at 3 and 7 days post-treatment. Ablated and non-ablated liver tissues were collected to measure HGF. Liver tissue was also collected and analyzed for periablational infiltrating cell populations, tumor proliferation indices, and growth factors and growth factor receptors linked to RFA-induced tumorigenesis [R]. Analyzed infiltrating cell populations included alpha-smooth muscle actin positive activated myofibroblast (α-SMA), macrophages (CD68), natural killer cells (KIR3DL1), and mature dendritic cells (CD80). Tumorigenesis markers included Ki-67 (cell proliferation) and CD34 (microvascular density). Growth factors included the receptor for vascular endothelium growth factor-1 (VEGF-R1), a key marker for HCC angiogenesis pathways, and fibroblast growth factor-19 (FGF-19), an HGF/c-MET/STAT3 dependent tumorigenic promoter of HCC.

Next, 8 BALB/c and 8 C57BL/6 non-tumor bearing mice were randomized into two treatment groups: 1) hepatic RFA alone, and 2) hepatic RFA with a single dose of atorvastatin. Animals were sacrificed at 7d post-ablation. Liver tissues were harvested and analyzed for myofibroblast infiltration (α-SMA), cell proliferation (Ki-67), and microvascular density (CD34).

## Phase II

Determination of the role of myofibroblasts in hepatic RFA tumorigenesis of macrometastasis. Here, 48 rats with established subcutaneous R3230 adenocarcinoma breast tumors were randomized into 4 treatment groups: 1) sham treatment, 2) sham with multiple doses of atorvastatin at 0, 24, and 48h, 3) hepatic RFA alone, and 4) hepatic RFA with multiple doses of atorvastatin. Animals were sacrificed at day7 post-treatment. Tumor growth curves were plotted and analyzed.

## Phase III

Role of myofibroblasts in implantation and tumorigenesis of micrometastases following hepatic RFA. Two murine models of diffuse intrahepatic colorectal cancer micrometastases were used to study the role of myofibroblasts in RFA induced micrometastatic tumor cell implantation and growth. C-MET–positive murine colorectal cancer cell lines, CT26 and MC38, in syngeneic mouse strains, BALB/c and C57BL/6 mice, respectively, were used [17]. Animals (n = 24 per strain) were randomly assigned to 4 treatment groups: 1) sham procedure followed by intrasplenic tumor cell injection 1d later, 2) sham followed by intrasplenic tumor cell injection and daily atorvastatin for 3d; 1d prior to implantation, day of implantation and 1d after to implantation, 3) hepatic RFA followed by intrasplenic tumor cell injection, and 4) hepatic RFA followed by intrasplenic tumor cell injection and daily atorvastatin. Animals were sacrificed at 14d post-RFA for the CT26/BALB/c model and 21d post-RFA for the MC38/C57BL/6 model. Whole livers were harvested to assess for tumor load, myofibroblast infiltration (α-SMA), cell proliferation (Ki-67), and microvascular density (CD34).

## Phase IV

Role of myofibroblasts in renal RFA-induced tumorigenesis of distant macrometastasis. To determine whether renal RFA-induced tumor growth is influenced by infiltrating

myofibroblasts, RFA of the kidney was performed on 32 Fischer rats implanted with subcutaneous R3230 breast adenocarcinoma were randomized into 4 treatment groups: 1) sham, 2) sham combined with multiple doses of atorvastatin, 3) renal RFA alone, and 4) renal RFA combined with multiple doses of atorvastatin. Animals were sacrificed at 3d and 7d after sham or RFA procedures. As described in phase I, animals were analyzed utilizing similar end points, including tumor growth curves, ELISA to measure HGF in the ablated kidney, IHC of α-SMA and CD68 in periablational tissue, and Ki67, CD34, and VEGF-R1 in periablational kidney and breast adenocarcinoma tumor tissues.

Further detailed descriptions of animal models used, animal surgery, RFA, anesthesia and pain alleviation, sacrifice, sample collection, proteomic analysis and statistical analysis can be found in the supplemental section, and have been previously described in detail [17, 34–37].

**Ethics statement.** All of the methods and studies reported were approved by our Institutional Animal Care and Use Committee. Specifically, rat studies were performed at Beth Israel Deaconess Medical Center, while mouse experiments were performed at Hadassah Hebrew University Medical Center in accordance with protocols approved by the respective Institutional Animal Care Committees. For rat studies, the maximum permissible diameters for R3230 breast adenocarcinomas, as per approved IACUC protocol, was a mean diameter of 2 cm. Tumor sizes were assessed using manual caliper measurements of two largest perpendicular dimensions and calculating mean diameters. After surgical procedures, the animals were monitored on a daily basis to determine any specific clinical signs of postoperative pain and distress. Specific signs of weakening or distress included, but were not limited to: total combined tumor burden greater than 2 cm or 10% of body weight, decreased appetite, reduced ambulation (interfering with or hampering with the animal's ability to obtain food and water, bear its own weight, or regain normal posture if placed on the back), weight loss (20% loss of body weight within 7 days, measured every 2-3d), tumor ulceration or necrosis for subcutaneous tumors. Post-surgically, if signs of distress were detected, a single dose of buprenorphine SR (72 hour duration of action) was administered at the site of surgery. In cases of tumor burden exceeding approved limits, animal were euthanized. For mouse experiments, general condition and animal well-being were used as metrics of tumor burden in lieu of a maximum permissible size due to microscopic nature of CRC liver deposits. Similar to rat experiments, post-operatively, animals were monitored on a daily basis to determine any specific clinical signs of postoperative pain and distress. For post-operative pain and distress, a single dose of buprenorphine SR was administered at the site of surgery. If signs of tumor burden overgrowth were detected, such as decreased appetite, reduced ambulation and or weight loss animals were euthanized.

## Results

### 1. Atorvastatin reduces RFA-induced myofibroblast periablational infiltration

Immunofluorescence assays of activated myofibroblasts in rat liver demonstrated significantly higher intensity of infiltrating α-SMA positive cells in the RFA treatment group as compared to sham or sham with atorvastatin. Specifically, activated myofibroblasts accumulated in the periablational rim, increasing significantly from 3 to 7 days (p-value <0.001, RFA and RFA plus atorvastatin). Atorvastatin following RFA reduced myofibroblasts by 50% at 3d and 7d versus RFA treatment alone. (8,437,720.1± 694416.5 vs. 821117.8 ± 338805.1 vs. 1039511.7 ± 247990.3 vs. 4193656.4 ± 260689.6 Pixel Intensity (PI) at 3d, p-value < 0.001 for all comparisons with RFA and p-value <0.001 comparing control groups to RFA and atorvastatin and 12324537.7 ± 1478667.1 vs. 827895.6 ± 495894.0 vs. 1483688.6 ± 886000.3 vs.

5967565.4 ± 878778.3 PI at 7d, p-value < 0.001 for all comparisons with RFA and p-value <0.001 comparing control groups to RFA and atorvastatin). (Fig 1A).

In BALB/c and C57BL/6 mouse models, similar findings were noted. Specifically, a significant reduction of periablational rim thickness of α-SMA positive cells was demonstrated when comparing hepatic RFA alone to RFA with atorvastatin (78.98 μm ± 30.1 vs 38.61 μm ± 21.2, for BALB/c mice (p <0.001) and 74.13 μm ± 28.07 vs 43.11 μm ± 19.80, for C57BL/6 mice) (Fig 1B).

## 2. Myofibroblast inhibition suppresses RFA-induced upregulation of tumor promoting cytokines

Samples of both periablational and non-ablated liver 3d and 7d after RFA treatment demonstrated elevated levels of HGF when compared to control groups and RFA plus atorvastatin (p-value < 0.05, all comparisons; Table 1). There was no statistically significant difference between RFA and atorvastatin compared to control groups (p-value > 0.05 for all comparisons) (Fig 2A). Additionally, periablational rat liver at 7d post-treatment demonstrated elevated FGF19 and VEGRFR-1 levels in the RFA arm compared to sham or sham plus atorvastatin (p-value <0.001, for all comparisons), which were reduced by 50% in RFA plus atorvastatin (p-value <0.001, vs. RFA and p-value <0.001 vs control groups).

## 3. Myofibroblast blockade eliminates hepatic RFA-induced tumorigenesis and proliferation of distant established tumors

RFA alone demonstrated significantly increased distant mean tumor diameter from 2d post-ablation to 7d post-ablation, when compared to sham, sham plus atorvastatin, or RFA plus atorvastatin (p-value <0.05, all comparisons). Additionally, RFA plus atorvastatin treatment did not show statistically significant different tumor growth when compared to sham or sham

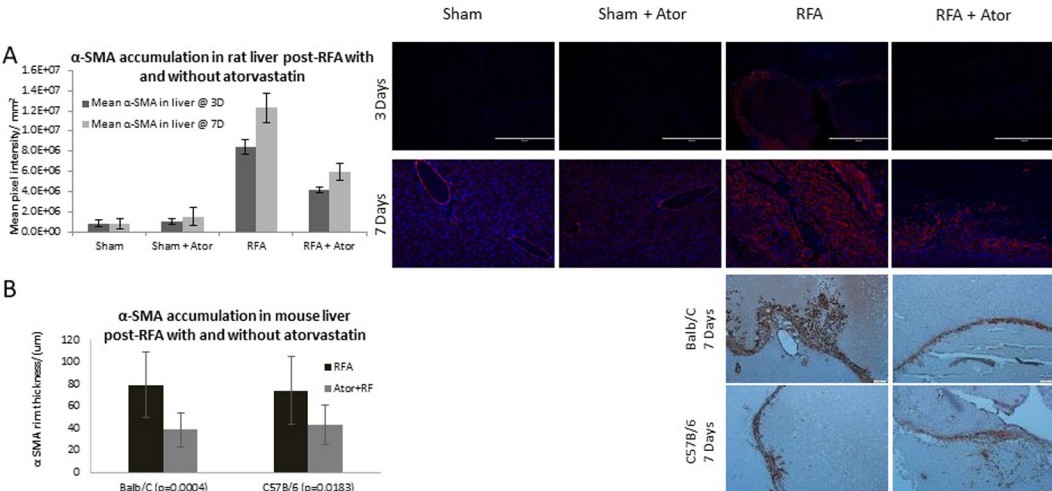

**Fig 1. Atorvastatin reduces RFA-induced myofibroblast periablational infiltration in treated liver in three rodent models.** Animal livers were evaluated using immunofluorescent staining. Statistically significant increased immunofluorescent intensity of infiltrating α-SMA positive cells was detected in the RFA treatment group as compared to sham or sham plus atorvastatin. Atorvastatin treatment significantly reduced α-SMA cell positivity at 3 and 7 days post-treatment (p-value <0.001, both comparisons), while remaining elevated to control groups at both time points (p-value <0.001) (A). Immunohistochemical analysis of BALB/c and C57B/6 mice (B) livers 7 days after RF ablation with and without atorvastatin treatment likewise demonstrated significant reduction of α-SMA periablational rim thickness in the combined RFA and atorvastatin group in both models (p-value <0.001 and p-value <0.05, respectively).

**Table 1. Effect of myofibroblast blockade on Hepatocyte growth factor upregulation in the periablational rim and distant liver post-RFA.**

|  |  | Sham | Sham + Ator | RFA | RFA + Ator |
|---|---|---|---|---|---|
| Periablational rim (OD ± SD) | 3D | 20 ± 3.2 | 15.5 ± 4.2 | 39.1 ± 6.7 | 19.7 ± 2 |
|  | 7D | 11.2 ± 3 | 7.2 ± 2 | 22.5 ± 2.8 | 7.3 ± 3.5 |
| Distant liver (OD ± SD) | 3D | 15.8 ± 3.5 | 20.7 ± 2.7 | 22.6 ± 3.9 | 9.4 ± 3.3 |
|  | 7D | 11.0 ± 2.4 | 10.6 ± 1.4 | 15.8 ± 3.0 | 9.7 ±1.2 |

plus atorvastatin (p-value >0.05, both comparisons) (Fig 3A). At 7d post-ablation, hepatic RFA alone demonstrated significantly increased macrophages (CD68), mature dendritic cells (CD80), and natural killer cells (KIR3DL1) in the periablational rim when compared to sham, RFA plus atorvastatin, or sham plus atorvastatin (p-value <0.001, for all comparisons) (Table 2). Similarly, increased cellular proliferation (Ki-67) and microvascular density (CD34) were observed in the periablational rim at 3d and 7d following hepatic RFA alone, which were successfully reduced to 50% for Ki67 (p-value <0.001, vs RFA plus atorvastatin and p-value <0.001 vs control groups) and to baseline levels for CD34 in RFA plus atorvastatin (p-value <0.001, all comparisons between RFA vs RF plus atorvastatin or control groups; p-value >0.05 comparing RF plus atorvastatin and control groups). (Fig 3B and 3C and Table 2).

## 4. Myofibroblast blockade eliminates hepatic RFA-induced tumorigenesis of micrometastatic implants

In MC38 and CT26 colorectal metastatic models, hepatic RFA alone increased the total number of tumor nodules, tumor area ratio, and nodule size compared to sham, sham plus

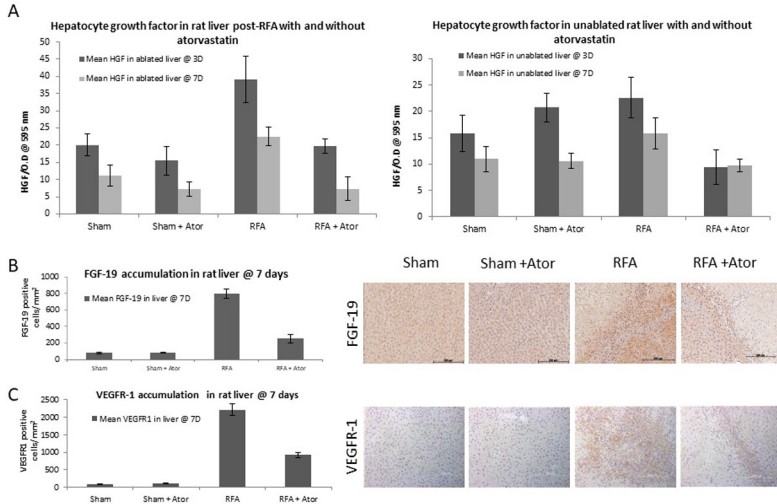

**Fig 2. Myofibroblast blockade inhibits RFA-induced upregulation of tumor promoting cytokines.** Ablated (and un-ablated, HGF only) livers were analyzed for known RF induced tumorigenic growth factors (HGF and VEGFR-1) as well as myofibroblast dependent growth tumorigenic growth factors (FGF-19). (A) Enzyme-Linked immunosorbent assays (ELISA) demonstrated statistically significant elevated levels of HGF in the RFA group when compared to control groups and RFA plus atorvastatin (p-value < 0.05 for all comparisons) in treated rat liver. No statistically significant difference was detected between RFA plus atorvastatin and control groups at 3 and 7 days (p-value > 0.1, all comparisons). ELISA also demonstrated statistically significantly elevated HGF levels in non-ablated rat liver with RFA treatment in comparison to control groups and combined atorvastatin with RFA at 3 and 7 days post-ablation (p-value < 0.05, all comparisons), with, no statistically significant difference detected between RFA plus atorvastatin treatment and control groups (p-value > 0.1, all comparisons). Significantly increased FGF-19 (B) and VEGRFR-1 (C) levels were demonstrated in RFA compared to sham or sham plus atorvastatin 7 day post-treatment (p-value <0.001, all comparisons) and were successfully reduced to half in RFA plus atorvastatin (p-value <0.001, all comparisons with RFA and p-value <0.001, for all comparisons to control groups).

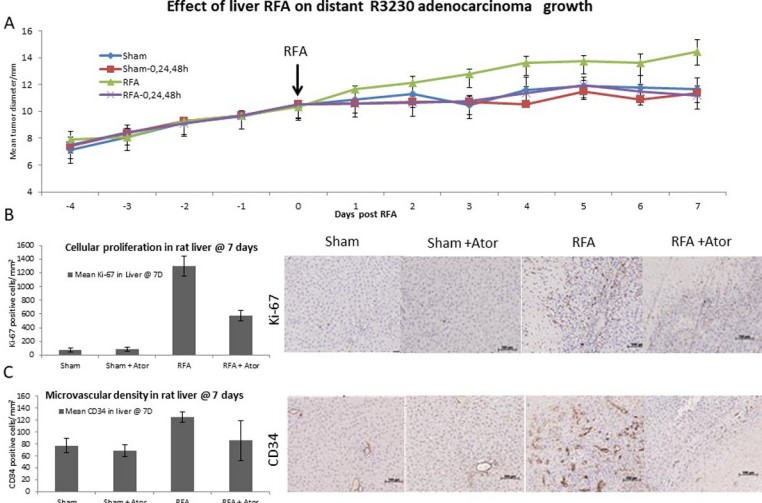

**Fig 3. Myofibroblast blockade eliminates hepatic RF driven tumorigenesis and of proliferation distant established tumors.** (A)Tumor growth mean diameter measurements demonstrate RFA treatment significantly increased the distant tumor diameter when compared to sham, sham plus atorvastatin and RFA plus atorvastatin starting 3 days after ablation (p-value <0.05, all comparisons). Furthermore, no statistically significant different tumor growth was demonstrated in RFA plus atorvastatin when compared to sham or sham plus atorvastatin (p-value >0.05, both comparisons). Immunohistochemical analysis (B&C) and comparison of cellular proliferative (Ki-67) and microvascular density (CD34) indices was performed. Significantly elevated Ki-67 (B) and CD34 (C) levels were detected in RFA as compared to sham and sham plus atorvastatin groups at 7 days, which were successfully reduced to the baseline in RFA plus atorvastatin for CD34 (p-value <0.001, all comparisons between RFA alone vs. RFA plus atorvastatin or control groups; p-value >0.05 comparing RFA plus atorvastatin and control groups) and to half for Ki-67 (p-value <0.001, all comparisons with RFA and p-value <0.001, all comparisons to control groups).

atorvastatin treatment and combined RFA with atorvastatin (p-value <0.05, all comparisons for MC38 and p-value <0.001, all comparisons for CT26). Combination RFA plus atorvastatin reduced tumor burden compared to RFA alone to baseline levels observed with sham or sham plus atorvastatin treatment (p-value >0.1 for all comparisons, both models) (Fig 4).

In the hepatic RFA arm, increased activated myofibroblasts were observed in distant intra-hepatic metastases at the time of sacrifice (MC38 and CT26 at 21 and 14 d) when compared to sham, sham plus atorvastatin, or RFA plus atorvastatin (p-value <0.001 for all comparisons). When combining RFA with atorvastatin, myofibroblast infiltration levels were reduced to baseline control groups' levels (p-value >0.05 for all comparisons).

Similarly, in animals treated with hepatic RFA alone, metastatic tumors demonstrated increased cellular proliferation (Ki-67) and microvascular density (CD34) compared to sham, sham plus atorvastatin, and RFA plus atorvastatin (p-value <0.001, all comparisons). However,

**Table 2. Effect of myofibroblast blockade on cellular trafficking, downstream tumorigenic growth factors and proliferation in rat livers' periablational rim collected 7 days post hepatic ablation.**

| | | **Sham** | **Sham + Ator** | **RFA** | **RFA + Ator** |
|---|---|---|---|---|---|
| Cellular infiltrates (PI/mm$^2$±SD) | CD68 | 179.2 ± 40.5 | 136.2 ± 38.1 | 3521.2 ± 297.3 | 2266.1 ± 123.4 |
| | CD80 | 11 ± 2 | 12 ± 6 | 325.2 ± 55.5 | 53 ± 18.2 |
| | KIR3DL3 | 1049.1 ± 111.2 | 937.0 ± 92.2 | 27955.9 ± 2572.7 | 13822.5 ± 1447.7 |
| Tumorigenic Growth Factors (PI/mm$^2$±SD) | FGF-19 | 81.2 ± 8.6 | 85.7 ± 7.0 | 795.7 ± 56.9 | 251.8 ± 48.0 |
| | VEGFR-1 | 94.4 ± 13.3 | 98.7 ± 10.9 | 2216 ± 158.7 | 918.7 ± 81.5 |
| Prolif. indices (PI/mm$^2$±SD) | Ki-67 | 76.5 ± 26.0 | 90.1 ± 27.4 | 1299.4± 140.5 | 571.9 ± 74.7 |
| | CD34 | 77.2 ±11.8 | 69 ± 10.3 | 125.7 ± 8.7 | 85.8 ± 33.4 |

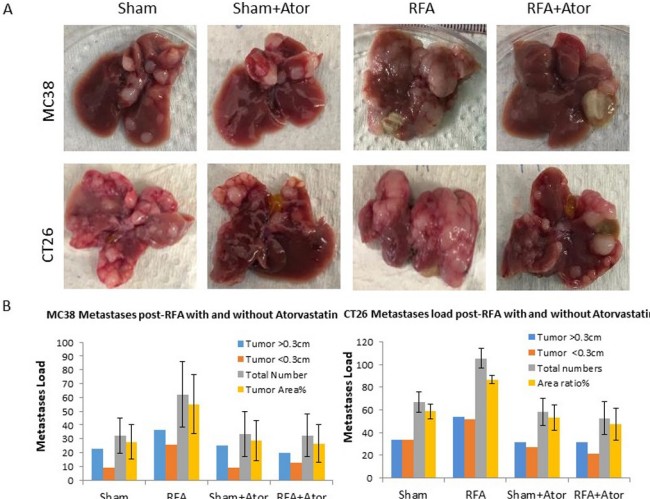

**Fig 4. Myofibroblast blockade eliminates hepatic RF driven tumorigenesis of micrometastatic implants.** RFA statistically significantly increased total number of tumor nodules, nodule size and tumor area ratio when compared to sham and sham plus multi-atorvastatin treatment and combination of RFA with atorvastatin in murine MC38 and CT26 colorectal metastatic models (p-value <0.05, all comparisons). Additionally, RFA plus atorvastatin treatment demonstrated statistically significant lower tumor burden compared to RFA, with no statistically significant increase in tumor burden noted when compared to sham or sham plus atorvastatin treatment (p-value >0.1 for all comparisons, both models).

there was no difference in the assayed markers when comparing sham, sham plus atorvastatin and RFA plus atorvastatin indicating successful reduction of cellular markers to baseline levels with atorvastatin treatment (p-value >0.05 all comparisons). (Fig 5 and Table 3).

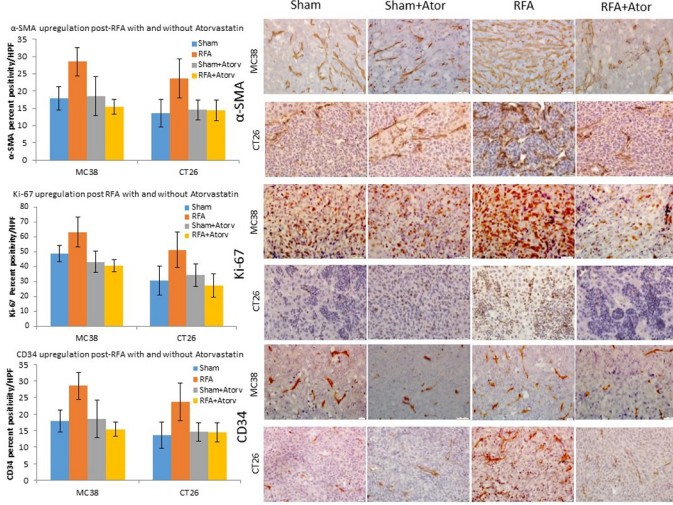

**Fig 5. Myofibroblast blockade suppresses RF induced tumor proliferation of micrometastatic implants in murine colorectal cancer models.** Immunohistochemical analysis of MC38 and CT26 demonstrated increased infiltration of activated myofibroblasts, increased cell proliferative indices and microvascular density in RFA groups when compared to sham, sham plus atorvastatin and RFA plus atorvastatin (p-value <0.001 for all comparisons, for both models). Myofibroblast infiltration levels were reduced to baseline control groups' levels in RFA with atorvastatin (p-value >0.05 for all comparisons, for both models). Likewise, RFA also demonstrated increased Ki-67 and CD34 signals in metastatic nodules compared to sham, sham plus atorvastatin and RFA with atorvastatin (p-value <0.001, all comparisons, both models), which were reduced to baseline levels in RFA plus atorvastatin treatment (p-value >0.05 all comparisons).

**Table 3. Effect of atorvastatin on myofibroblast infiltration and proliferation in CRC metastasis after hepatic RF.**

| | | Sham | Sham + Ator | RFA | RFA + Ator |
|---|---|---|---|---|---|
| Cellular infiltrates | | | | | |
| α-SMA (%±SD) | MC38 | 7.1 ± 1.7 | 6.1 ± 1.8 | 19.6 ± 6.5 | 7.6 ± 1.6 |
| | CT26 | 5.3 ± 1.6 | 7.9 ± 2.4 | 15.7 ± 5.1 | 6.6 ± 2.2 |
| Proliferative indices | | | | | |
| Ki-67 (%±SD) | MC38 | 48.4 ± 10 | 43.2 ± 7.6 | 63.2 ± 11.7 | 40.7 ± 7.9 |
| | CT26 | 30.9 ± 5.6 | 34.4 ± 7.3 | 51.5 ± 9.9 | 27.4 ± 4 |
| CD34 (%±SD) | MC38 | 17.9 ± 3.4 | 18.5 ± 5.6 | 28.5 ± 4.1 | 15.4 ± 2.2 |
| | CT26 | 13.6 ± 4 | 14.6 ± 2.9 | 23.7 ± 5.6 | 14.5 ± 3 |

## 6. Atorvastatin partially suppresses RFA-induced periablational myofibroblast infiltration in ablated kidney

Following renal RFA alone, increased periablational infiltration of activated myofibroblasts (α-SMA) (S1 Fig in S1 File) and macrophages (CD68) were observed compared to control groups and RFA with atorvastatin (p <0.05, all comparisons). However, adjuvant atorvastatin following renal RFA did not reduce infiltrating cells to baseline levels (p <0.05, all comparisons to control groups). (S1 Table in S1 File).

## 7. Myofibroblast blockade reduces renal RFA-induced tumor promoting cytokines, tumorigenesis of distant established tumors, and periablational and distant tumor proliferation

Renal RFA alone upregulated HGF at 3d and 7d compared to sham and sham plus atorvastatin (p <0.05 for all comparisons). Adjuvant atorvastatin reduced post-RFA HGF in periablational kidney compared to RFA alone (p <0.05, all comparisons). Compared to hepatic RFA, combination atorvastatin following renal RFA did not return completely to baseline, with HGF levels still elevated compared to sham treatment (p <0.05 for all comparisons) (S2A Fig and S1 Table in S1 File). VEGFR-1 level was also increased in both the renal periablational rim and distant tumor at 3d and 7d compared to sham, sham plus atorvastatin, or RFA plus atorvastatin (p <0.05, all comparisons). RFA plus atorvastatin demonstrating higher VEGFR-1 levels than control groups ((p <0.05, all comparisons) (S2B and S2C Fig and S1 Table in S1 File).

Renal RFA alone increased mean tumor diameter of the distant subcutaneous breast tumor from 3d to 7d post-ablation compared to sham, sham plus atorvastatin, or RFA plus atorvastatin (p <0.05, all comparisons). RFA plus atorvastatin treatment did not increase of mean tumor diameter when compared to control groups (p-value >0.05, all comparisons) (S3 Fig in S1 File).

Similarly, renal RFA alone increased periablational and distant tumor cellular proliferation and microvascular density compared to control groups. Atorvastatin partially attenuated increases in proliferation and microvascular density when combined with RFA (p <0.05, all comparisons). (S4 Fig and S1 Table in S1 File).

## Discussion

There is a growing body of clinical reports of accelerated tumor growth after hepatic and renal RFA [18, 19, 34, 38] of primary and metastatic tumors [38, 39]. Studies investigating this phenomenon have revealed temporally-associated cytokine surges following liver RFA, specifically, interleukin-6 [28], a key pro-inflammatory marker associated with HGF/c-MET/STAT3

pathway upregulation, which is a known key pathway in tumorigenesis. Preclinical models have also correlated RFA-induced increases of inflammatory cytokines IL-6 [27], HGF [17], and VGEF [15], to upregulation of downstream key proliferative pathways [14], and resultant accelerated distant tumor growth peaking 6h to 3d following ablation [16]. At the same time, animal studies have demonstrated increased infiltrating immune cells in the periablational rim, namely polymorphic neutrophils, macrophages, and activated myofibroblasts [27]. Fibroblasts, specifically activated α-SMA–positive myofibroblasts, have been implicated as a crucial subset of the tumor microenvironment promoting gastrointestinal carcinomas [40, 41] promoting tumor growth [31], an immunosuppressive immunity [42], and are associated with chemo-resistance [43]. Myofibroblasts are also known as major producers of HGF [43] and interleukin-6 [44], key cytokines implicated in post-ablation tumorigenesis. Accordingly, in the current study we attempt to elucidate the role of myofibroblasts in the upregulation of the HGF/c-MET/STAT3 pathway and overall tumorigenesis following radiofrequency ablation.

Prior studies have shown that undesirable pro-tumorigenic effects can be successfully suppressed by selective inhibition of interleukin-6 [34], and c-MET, or STAT3 [15–17] in animal models. We postulated and demonstrate here that a similar approach can potentially be sought to modulate cell populations that orchestrate tumor growth and progression, such as myofibroblasts. Accordingly we tested the use of atorvastatin, an established lipid lowering medication, which has known inhibitory effects on myofibroblasts [45], in the post ablation setting. Through inhibition of small GTPases (RhoA and Ras) atorvastatin has demonstrated portal pressure lowering effects and attenuated fibrosis in preclinical models [46]. Additionally, atorvastatin has been shown in large retrospective studies to be associated with improved survival in non-small lung cancer [47] and prostate cancer [48], albeit the exact mechanism remains to be determined.

In the first phase of this study, increased myofibroblast infiltration in the periablational rim post-RFA was again demonstrated, but also now successfully attenuated by atorvastatin treatment in two different animal models. Furthermore, we demonstrate that atorvastatin markedly attenuates a host of post-RFA tumorigenesis-inducing phenomenon including release of HGF and key dependent growth factors such as FGF-19 and VEGFR-1, as well as tumor markers such as Ki-67 and CD34 implicating fibroblast proliferation in a cascade that leads to proliferation and angiogenesis. Interestingly, this is also associated with a decrease in other infiltrating cell populations observed which can be attributed to myofibroblast blockade, a known key regulator of the inflammatory cellular infiltrate [49]. Specifically, we report significant modulation of macrophages, mature dendritic cells, and natural killer cells. This underscores the complex interactions between infiltrating cells that ultimately govern tumorigenesis. We therefore hypothesize that despite selective myofibroblast inhibition a much broader cohort of cells are impacted due to the complex interdependence and regulatory roles between infiltrating periablational cells. Alternatively or simultaneously, atorvastatin may have a broad inhibitory effect on multiple inflammatory cells. Furthermore, this observation makes a strong case for cell specific targeting strategies, especially, of other abundant cell populations with known tumor promoting properties within the tumor microenvironment such as M2 macrophages.

Our subsequent tumor growth studies were in concert with immunohistochemical and proteomic findings. In a rat model, liver RF induced accelerated distant tumor growth which was successfully returned to baseline growth rates by a multi-day course of atorvastatin treatment. Similar effects were reproduced and confirmed in a second mouse colorectal model that simulates intrahepatic micrometastatic implantation from portal shower. Here too, atorvastatin treatment demonstrated successful attenuation of RF driven increased number of micrometastatic implants, nodule size, and overall hepatic tumor burden.

Finally, we confirmed similar findings for a second site in which RFA is commonly practiced. Specifically, we not only confirmed that renal ablation accelerates distant subcutaneous tumor growth [34], but also that this phenomenon can be suppressed, albeit only partially, by atorvastatin treatment. This suggests that RF induced distant tumorigenesis is not an isolated hepatic phenomenon and can most likely be observed with the ablation of any organ. It also reaffirms that organ specific interactions are at play after inflecting thermal injury upon it. For instance, atorvastatin treatment in liver RF models successfully attenuated any RF-driven HGF upregulation. While in kidney RF models, myofibroblast blockade significantly reduced HGF levels below RF groups but not to control groups. This highlights the potential role of organ specific cells and growth pathways in post ablation tumorigenesis.

Potential limitations of our study include those based upon experimental design. For example, we acknowledge that the subcutaneous implantation models used to ablate primary organs (liver or kidney) and study effects on a distant tumor of a separate origin (breast) represent an uncommon clinical scenario. Nevertheless, such a model can strengthen the argument that the observed phenomena transcend tumor origin and cellular homogeneity of the primary tumor and affected satellites. Likewise, our complementary intrasplenic injection model of tumor implantation via trans-splenic injection may create an artificial potentially greater portal cellular load than in a typical clinical scenario. Additionally, the timing of RF in relation to the portal tumor cell delivery does not necessarily reflect a longer tumor dwelling time in most clinical scenarios. Thus, although the tumor models used are well-characterized allowing comparison with other preclinical studies of RF ablation, careful interpretation and application of the results is warranted. Indeed, atorvastatin specific effects described in these models may vary in other models based on tissue susceptibility.

In conclusion, while radiofrequency ablation has demonstrated robust clinical efficacy in treatment of intrahepatic and renal tumors, the inadvertent ablation of normal parenchyma of the primary organ can stimulate distant tumor growth and proliferation. Our study offers three clinically pertinent insights. Activated myofibroblasts are a key orchestrating cell population in post RF ablation distant tumor growth, primarily through induction of the HGF/c-MET/STAT3 axis. Post RFA distant tumor growth is a phenomenon that transcends phenotype of ablated primary organ ablated (including liver or kidney) and phenotype of tumor (colorectal cancer or breast cancer) or size (micro or macrometastatic tumors). Finally, selective inhibition of RF induced distant tumor growth can be achieved with by targeted cellular inhibition of activated myofibroblasts. Mouse colorectal models demonstrated reduction of cellular, growth signal and tumor burden markers to baseline levels when combining atorvastatin with RF. However, in rat models, although tumor growth studies demonstrated complete negation of RF induced distant tumor growth with atorvastatin, cellular and growth signals were only partially curbed. Thus, further evaluation, especially, including long-term survival studies and combined targeted inhibition of key cellular and cytokine agents of established tumorigenesis pathways.

## Supporting information

**S1 File.**
(DOCX)

## Author Contributions

**Conceptualization:** S. Nahum Goldberg, Muneeb Ahmed.

**Data curation:** Marwan Moussa, David Mwin, Eithan Galun.

**Formal analysis:** Marwan Moussa, David Mwin, Haixing Liao, M. Fatih Atac, S. Nahum Goldberg, Muneeb Ahmed.

**Funding acquisition:** S. Nahum Goldberg, Muneeb Ahmed.

**Investigation:** David Mwin, Haixing Liao, Aurelia Markezana, Eithan Galun.

**Methodology:** David Mwin, Haixing Liao, Aurelia Markezana, Eithan Galun, S. Nahum Goldberg, Muneeb Ahmed.

**Project administration:** David Mwin, Haixing Liao, Aurelia Markezana.

**Supervision:** S. Nahum Goldberg, Muneeb Ahmed.

**Validation:** Marwan Moussa, S. Nahum Goldberg, Muneeb Ahmed.

**Writing – original draft:** Marwan Moussa, David Mwin.

**Writing – review & editing:** Marwan Moussa, M. Fatih Atac, S. Nahum Goldberg, Muneeb Ahmed.

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
