## [Decision Letter · Decision Letter 0]

7 Jan 2022

PONE-D-21-21027

Myofibroblasts: A key promoter of tumorigenesis following radiofrequency tumor ablation.

PLOS ONE

Dear Dr. Ahmed,

Thank you for submitting your manuscript to PLOS ONE. After careful consideration, we feel that it has merit but does not fully meet PLOS ONE’s publication criteria as it currently stands. Therefore, we invite you to submit a revised version of the manuscript that addresses the points raised during the review process.

We look forward to receiving your revised manuscript.

Kind regards,

Hanna Landenmark

Senior Editor, PLOS ONE

on behalf of

Yanlin Yu

Journal Requirements:

1) Please include a complete ethics statement in the Methods section. Please ensure that you state for each experiment conducted the name of the ethics committee that approved the study, the method of euthanasia, the method of anesthesia and any efforts to alleviate suffering. Please also provide for each experiment the following information: 1. maximum sizes tumors grew, 2. how tumor size was calculated, 3. the post-operative and post tumor implantation care received by the animals, including the frequency of monitoring and the criteria used to assess animal health and well-being. Please also state whether the IACUC provided approval for any large tumor sizes. 

3) Please note that concerns were raised about image quality, however, in the event of publication, all images will be uploaded in the original resolution rather than that seen in the PDF, so there is no need to enhance your figures if the original resolution is sufficiently clear.

Additional Editor Comments (if provided):

The authors report the role of activated myofibroblasts in RFA-induced tumorigenesis using murine models. The study is well designed, the results are interesting. The manuscript is well written, but the images of the figures are not clear. So, I suggest publishing the vision of the manuscript with clear figures images.

Reviewers' comments:

Reviewer's Responses to Questions

**Comments to the Author**

1. Is the manuscript technically sound, and do the data support the conclusions?

Reviewer #1: Yes

2. Has the statistical analysis been performed appropriately and rigorously? 

Reviewer #1: Yes

3. Have the authors made all data underlying the findings in their manuscript fully available?

Reviewer #1: Yes

4. Is the manuscript presented in an intelligible fashion and written in standard English?

Reviewer #1: Yes

5. Review Comments to the Author

Reviewer #1: RF ablation was reported to increase growth of distant untreated tumor foci following local tumor ablation. Thus the researchers were looking to demonstrate that undesirable pro-tumorigenic effects induced by RF ablation, is carried out by myofibroblasts and that their suppression might affect tumor growth and progression.

They report on the key role of activated myofibroblasts in RFA-induced distant metastases growth and successful pharmacologic blockade in mouse and rat models of colon, renal and breast experimental models.

Activated myofibroblasts are a key cell population in post RF ablation distant tumor growth, primarily through induction of the HGF/c-MET/STAT3 axis. Post RFA distant tumor growth was observed in different phenotypes of ablated primary organ such as liver or kidney or phenotype of tumors such as colorectal cancer or breast cancer. They also observed that RF induced distant tumor growth can be achieved by inhibition of activated myofibroblasts using atorvastatin, an established lipid lowering medication, and anti-fibroblast agent.

This is a well executed research with interesting and important results.

I suggest that the authors will discuss the possibility that macrophage inhibition will achieve similar effects?

6. PLOS authors have the option to publish the peer review history of their article (what does this mean?). If published, this will include your full peer review and any attached files.

Reviewer #1: **Yes: **Yona Keisari

---

## [Author Response · Author response to Decision Letter 0]

20 Jan 2022

Dear Drs. Landemark and Yu:

Thank you very much for your favorable review of our submitted original manuscript, PONE-D-21-21027, entitled “Myofibroblasts: A key promoter of tumorigenesis following radiofrequency tumor ablation”. Enclosed you will find our revised manuscript in which we have made the necessary requested changes. We will now detail the changes made in response to all comments on a point-by-point basis. 

Journal Requirements:

1. Please include a complete ethics statement in the Methods section. Please ensure that you state for each experiment conducted the name of the ethics committee that approved the study, the method of euthanasia, the method of anesthesia and any efforts to alleviate suffering. Please also provide for each experiment the following information: 1. maximum sizes tumors grew, 2. how tumor size was calculated, 3. the post-operative and post tumor implantation care received by the animals, including the frequency of monitoring and the criteria used to assess animal health and well-being. Please also state whether the IACUC provided approval for any large tumor sizes. 

Response: We now include a complete ethics statement in the Methods section. All of the methods and studies reported were approved by our Institutional Animal Care and Use Committee. We now include the additional information requested, including maximum tumor sizes (all of which were within the approved tumor size limit in our study protocol), how these were calculated, and the post-operative and post-tumor implantation care received. 

“All of the methods and studies reported were approved by our Institutional Animal Care and Use Committee. Specifically, rat studies were performed at Beth Israel Deaconess Medical Center, while mouse experiments were performed at Hadassah Hebrew University Medical Center in accordance with protocols approved by the respective Institutional Animal Care Committees. For rat studies, the maximum permissible diameters for R3230 breast adenocarcninomas, as per approved IACUC protocol, was a mean diameter of 2 cm. Tumor sizes were assessed using manual caliper measurements of two largest perpendicular dimensions and calculating mean diameters. After surgical procedures, the animals were monitored on a daily basis to determine any specific clinical signs of postoperative pain and distress. Specific signs of weakening or distress included, but were not limited to: total combined tumor burden greater than 2 cm or 10% of body weight, decreased appetite, reduced ambulation (interfering with or hampering with the animal's ability to obtain food and water, bear its own weight, or regain normal posture if placed on the back), weight loss (20% loss of body weight within 7 days, measured every 2-3d), tumor ulceration or necrosis for subcutaneous tumors. Post-surgically, if signs of distress were detected, a single dose of buprenorphine SR (72 hour duration of action) was administered at the site of surgery. In cases of tumor burden exceeding approved limits, animal were euthanized. For mouse experiments, general condition and animal well-being were used as metrics of tumor burden in lieu of a maximum permissible size due to microscopic nature of CRC liver deposits. Similar to rat experiments, post-operatively, animals were monitored on a daily basis to determine any specific clinical signs of postoperative pain and distress. For post-operative pain and distress, a single dose of buprenorphine SR was administered at the site of surgery. If signs of tumor burden overgrowth were detected, such as decreased appetite, reduced ambulation and or weight loss animals were euthanized.”

Response: There are no changes in the reference list.

3. Please note that concerns were raised about image quality, however, in the event of publication, all images will be uploaded in the original resolution rather than that seen in the PDF, so there is no need to enhance your figures if the original resolution is sufficiently clear.

Response: We have now uploaded images of higher resolution. We are happy to work with the editorial staff on these as required.

Additional Editor Comments:

The authors report the role of activated myofibroblasts in RFA-induced tumorigenesis using murine models. The study is well designed, the results are interesting. The manuscript is well written, but the images of the figures are not clear. So, I suggest publishing the vision of the manuscript with clear figures images.

Response: See response above.

Reviewer #1 - Specific comments:

“RF ablation was reported to increase growth of distant untreated tumor foci following local tumor ablation. Thus the researchers were looking to demonstrate that undesirable pro-tumorigenic effects induced by RF ablation, is carried out by myofibroblasts and that their suppression might affect tumor growth and progression.

They report on the key role of activated myofibroblasts in RFA-induced distant metastases growth and successful pharmacologic blockade in mouse and rat models of colon, renal and breast experimental models.

Activated myofibroblasts are a key cell population in post RF ablation distant tumor growth, primarily through induction of the HGF/c-MET/STAT3 axis. Post RFA distant tumor growth was observed in different phenotypes of ablated primary organ such as liver or kidney or phenotype of tumors such as colorectal cancer or breast cancer. They also observed that RF induced distant tumor growth can be achieved by inhibition of activated myofibroblasts using atorvastatin, an established lipid lowering medication, and anti-fibroblast agent.

This is a well-executed research with interesting and important results.

I suggest that the authors will discuss the possibility that macrophage inhibition will achieve similar effects?”

Response: We appreciate the Reviewer’s comments and enthusiasm for our work. We now address the potential role of macrophage suppression as well in the Discussion.

“Furthermore, this observation makes a strong case for cell specific targeting strategies, especially, of other abundant cell populations with known tumor promoting properties within the tumor microenvironment such as M2 macrophages.”

We have uploaded clean and annotated versions of the manuscript with tracked changes and figures, as requested. 

Once again we thank you for a favorable reply.

Sincerely,

Muneeb Ahmed, M.D. on behalf of all authors

---

## [Editor Report · Decision Letter 1]

23 Mar 2022

Myofibroblasts: A key promoter of tumorigenesis following radiofrequency tumor ablation.

PONE-D-21-21027R1

Dear Dr. Ahmed,

We’re pleased to inform you that your manuscript has been judged scientifically suitable for publication and will be formally accepted for publication once it meets all outstanding technical requirements.

Kind regards,

Yanlin Yu

Academic Editor

PLOS ONE

Additional Editor Comments (optional):

Dear Dr. Ahmed,

Thank you for submitting your revised manuscript to PLOS ONE. I am delighted to inform you that your manuscript, " Myofibroblasts: A key promoter of tumorigenesis following radiofrequency tumor ablation", has now been accepted for publication in PLOS ONE.

Kind regards,

Yanlin
---

## [Editor Report · Acceptance letter]

24 May 2022

PONE-D-21-21027R1 

Myofibroblasts: A key promoter of tumorigenesis following radiofrequency tumor ablation. 

Dear Dr. Ahmed:

I'm pleased to inform you that your manuscript has been deemed suitable for publication in PLOS ONE. Congratulations! Your manuscript is now with our production department. 

Kind regards, 

on behalf of

Dr. Yanlin Yu 

Academic Editor

PLOS ONE